# Whole body periodic acceleration in normal and reduced mucociliary clearance of conscious sheep

**Juan R. Sabater**[1☯], **Marvin A. Sackner**[1,2☯], **Jose A. Adams**[3☯]*, **William M. Abraham**[1☯†]

**1** Department of Research, Mount Sinai Medical Center, Miami Beach, Florida, United States of America,
**2** Department of Medicine, Mount Sinai Medical Center, Miami Beach, Florida, United States of America,
**3** Division of Neonatology, Mount Sinai Medical Center, Miami Beach, Florida, United States of America

☯ These authors contributed equally to this work.
† Deceased.
* tony-adams@msmc.com

**Data Availability Statement:** All relevant data are within the paper and its Supporting Information files.

## Abstract

The purpose of this investigation was to ascertain whether nitric oxide (NO) released into the circulation by a noninvasive technology called whole body periodic acceleration (WBPA) could increase mucociliary clearance (MCC). It was based on observations by others that nitric oxide donor drugs increase ciliary beat frequency of nasal epithelium without increasing mucociliary clearance. Tracheal mucous velocity (TMV), a reflection of MCC, was measured in sheep after 1-hour treatment of WBPA and repeated after pretreatment with the NO synthase inhibitor, L-NAME to demonstrated action of NO. Aerosolized human neutrophil elastase (HNE) was administered to sheep to suppress TMV as might occur in cystic fibrosis and other inflammatory lung diseases. WBPA increased TMV to a peak of 136% of baseline 1h after intervention, an effect blocked by L-NAME. HNE reduced TMV to 55% of baseline but slowing was reversed by WBPA, protection lost in the presence of L-NAME. NO released into the circulation from eNOS by WBPA can acutely access airway epithelium for improving MCC slowed in cystic fibrosis and other inflammatory lung diseases as a means of enhancing host defense against pathogens.

## 1.0 Introduction

Airway inflammation occurs in asthma, chronic obstructive pulmonary disease (COPD) and cystic fibrosis (CF). Although inflammation is most commonly linked to bronchoconstriction and airway hyper-responsiveness, mucociliary clearance (MCC) may also be diminished. Evidence supporting this comes from clinical observations impaired MCC during exacerbations of asthma [1, 2], COPD [3] and CF [4] as well as experimental data that inhaled inflammatory mediators, such as neutrophil elastase slows whole lung MCC and tracheal mucus velocity (TMV). [5, 6] TMV reflects changes in whole lung clearance measured with radiolabeled human serum albumin [7, 8] Effective mucus transport depends on the coordinated relationship among ciliated surface epithelium, the mucous (gel) layer, and the periciliary fluid (sol)

**Funding:** This study was funded via internal research funds from the Department of Research and Division of Neonatology.

**Competing interests:** The following authors JRS and WMA have declared that no competing interests exist. JAA owns minimal number of stocks in Noninvasive Monitoring Systems (NIMS), a company which manufactures a platform similar to the one described in this study, and co-patent USPATENT 9,622,933 – April 18, 2017; Passive Simulated Jogging Device (Gentle Jogger). MAS is president of Sackner Wellness Products, a company which manufactures a wellness device called Gentle Jogger referenced in the manuscript, and co-patent owner of the above mentioned patent. This does not alter our adherence to PLOS ONE policies on sharing data and materials.

layer. [9] Inflammatory mediators can disrupt this process predisposing to mucus plugging, infection, and decreased pulmonary function.

It is important to determine means to block and/or reverse slowed MCC. Administration of aerosolized neutrophil elastase (HNE), an inflammatory mediator that contributes to impaired mucus clearance in asthma, COPD, and CF can serve as a model to determine effectiveness. [5] Elastase is a known mucus secretagogue, [10] that has cilio-inhibitory properties [11] and by stimulating epithelial sodium channels can reduce periciliary fluid layer therby contributing to mucus stasis [12]. These collective actions of elastase on the various components of mucociliary function are consistent with our *in vivo* observations that inhaled elastase reduces MCC and TMV for prolonged periods. [5] Further, such effects can be prevented and reversed with natural and synthetic elastase inhibitors, including α1-protease inhibitor, [13] secretory leukocyte protease inhibitor, [14] and synthetic human neutrophil elastase (HNE) inhibitor [13] as well as beta-adrenergic agonists [5]. The latter appear to act through a NO pathway. [15]

Whole Body Periodic Acceleration (WBPA) which utilizes a motion platform to apply repetitive, sinusoidal, head to foot motion to the horizontally positioned body stimulates eNOS for release of NO into the circulation in humans and animal models through increased shear stress to the vascular endothelium. [16–18] In sheep, this technology modulated antigen induced inflammatory responses such as inhibiting nuclear factor kappa beta activity. [19] The purpose of the current investigation was to determine whether NO delivered via the blood stream could access airway epithelium to increase TMV and provide protection against human neutrophil elastase (HNE) induced slowed MCC. [20, 21]

## 2.0 Methods

### 2.1 Animal Preparation

All procedures used in this study were approved by the Mount Sinai Animal Research Committee, which is responsible for ensuring humane care and use of experimental animals, under protocol number 17-22-A-03. The study complies with the ARRIVE Guidelines for in vivo animal research reporting. (S1 Table) Five adult ewes weighing 20–40 kg were restrained in an upright position in a specialized body harness and placed within a modified shopping cart with heads immobilized. Nasal intubation was carried out with a cuffed endotracheal tube (ETT, ID = 7.5 mm; Mallinckrodt Medical Inc, St. Louis, MO). The cuff of the tube was placed just below the vocal cords and position verified with a flexible bronchoscope. After intubation, the animals were allowed to equilibrate for a period of 30 min before TMV measurements began. To minimize the possible impairment of TMV caused by inflation of the ETT cuff, a deflated cuff tube was utilized throughout the study except for the short periods of HNE challenge. [22] Inspired air was warmed and humidified with an ultrasonic nebulizer (Ultra-neb 99, The Devilbiss Co, Somerset, PA) during the treatment with WBPA, and a Bennett Humidifier (Puritan-Bennett, Lenexa, KS), the rest of the time between TMV measurements.

### 2.2 Measurement of TMV

This method has been published previously by our laboratory and involves the use of Teflon particles (~1 mm in diameter, 0.8 mm thick, and weight from 1.5 to 2 mg) introduced via a modified suction catheter into to the ETT. TMV was measured from the video recordings of the disks velocities. [5] [23]

## 2.3 Administration of aerosols

The administration of aerosols has also been previously reported by our laboratory in detail. This involves a jet nebulizer (Raindrop Nebulizer, Puritan-Bennett, Carlsbad, CA), with a dosimeter system and ventilator.[5, 23]

## 2.4 Reagents

Human neutrophil elastase (HNE; Elastin Products, Owensville, MO) was diluted in 3 ml of phosphate-buffered saline (PBS; pH 7.4) to a concentration of 0.1 units/ml and completely delivered as an aerosol (20 breaths/min). N-Nitro-L-arginine methyl ester hydrochloride (L-NAME; Sigma-Aldrich, St. Louis, MO) was given as an intravenous injection; the dose per animal was 25 mg/kg in 20 ml of 0.9% NaCl as previously described. [19]

## 2.5 Motion platform

Whole-body periodic acceleration (WBPA, aka pGz) was administered with a motion platform that was adapted to support a sheep restrained within the cart. The animal in cart was secured to the platform for the designated treatment times depending on the specific protocol. Acceleration parameters for all studies were set to 120 cpm and Gz of ±0.2. [19] For control studies, the animals in carts were placed on the platform for the appropriate time without motion.

## 2.6 Protocol

The series of experiments described below were conducted in the same group of 5 sheep separated by approximately one week for each series of investigations. In these experiments, a baseline TMV measurement was obtained 30 min after intubation.

In the first series of experiments, the effect of WBPA was evaluated on resting (basal) TMV and then determined if the stimulatory effects were related to production of NO. Baseline TMV was obtained and the animals were treated with WBPA for 1 hour. TMV was measured at 0.5, 1 and 2 h after stopping WBPA. This protocol was repeated in the presence of the NO inhibitor, L-NAME. After obtaining baseline TMV, L-NAME (25 mg/kg, i.v.) was administered and after 30 min the sheep were treated with WBPA for 1 h. Serial TMV measurements were then obtained at 0.5, 1 and 2 h after stopping WBPA. Finally, we studied the time course of the TMV response to L-NAME alone. For these experiments, a baseline TMV was obtained and then the sheep were given L-NAME and TMV was measured at 0.5, 1, 2, and 3h after L-NAME administration.

The second series of experiments were carried out to determine if WBPA could reverse HNE-induced slowing of TMV. This was followed by an investigation as to whether reversal of the HNE-induced slowing of TMV could be blocked by L-NAME. For the initial studies, baseline TMV was obtained and then the animals were challenged with an aerosolized dose of HNE (0.3 units). TMV was serially measured, from 60 to 90 minutes after HNE challenge until TMV decreased to at least 40% of baseline. Once this reduction in TMV was achieved, treatment with WBPA for 1 hour was initiated. Serial TMV measurements were obtained at 0.5, 1, 2, 3 and 4 h after stopping WBPA.

The effect of L-NAME (25 mg/kg) on the WBPA reversal of HNE-induced depression of TMV was determined after TMV had decreased by 40%. Treatment with L-NAME was then followed by a 1h treatment with WBPA. TMV measurements were serially obtained at 0.5, 1, 2, 3 and 4 h after stopping WBPA. At the conclusion of all experiments, the animals were returned to their herd. None of the experiments involved terminal endpoints.

**Table 1. Baseline TMV for the 5 series of studies.**

|  | WBPA | Elastase + WBPA | L-NAME + WBPA | 2Elastase + L-NAME + WBPA | L-NAME |
|---|---|---|---|---|---|
| Mean ± SE | 9.4±0.5 | 9.5±0.5 | 11.3±1.4 | 10.6±0.7 | 10.6±1.5 |
| N | 5 | 5 | 5 | 5 | 5 |

Values are means ± SE in mm/min. n, number of sheep. TMV, tracheal mucus velocity

## 2.7 Statistics

Statistical analysis was performed by using a commercially available program (SigmaStat for Windows, version 2.03; SPSS Inc, Chicago, IL). Comparisons of baseline TMV measurements were made with Kruskal-Wallis analysis of variance on ranks. For each experiment or trial (within-group analysis), data were analyzed across time, using one-way ANOVA for repeated measurements. If the null hypothesis was rejected, pairwise comparisons were made by using Tukey's test for multiple comparisons. Comparisons of experiments at specific time intervals were evaluated by using a t-test for two samples. A value of $P \leq 0.05$ was considered significant, using two-tailed analysis. All values in the text and figures are reported as means ± SE.

## 3.0 Results

The baseline TMV values for the different experiments are listed in Table 1. The baseline values for the start of each series of experiments did not statistically differ from each other.

Fig 1 depicts the changes in TMV after treatment with WBPA alone and when L-NAME was given prior to WBPA. Within 0.5h after stopping WBPA, TMV increased to 114 ± 4% above baseline and TMV continued to increase to a maximum of 136 ± 9% 1h after treatment. This increase in TMV persisted until the end of the 2h measurement. When L-NAME was given prior to WBPA, the stimulatory effect was completely blocked at all times (p<0.05). As depicted in Fig 1, TMV values after WBPA in the presence of L-NAME remained below baseline.

Fig 2 shows the TMV response to L-NAME alone. L-NAME caused an immediate reduction in TMV 30 minutes post-injection (67 ± 13%; p < 0.05 vs. baseline). TMV then returned toward baseline values. Although the 1h and 2h mean values were below the initial starting TMV, these values did not differ statistically from the baseline values. Three hours after L-NAME. TMV was significantly lower than baseline.

Fig 3 shows the effects of WBPA without and with L-NAME on the HNE-induced reduction in TMV. As seen in aforementioned studies, administration of HNE slowed TMV which reached similar levels in the two experimental groups (WBPA = 55 ± 2% vs. L-NAME + WBPA 50 ± 3%; p>0.05). Treatment with WBPA reversed the HNE-induced TMV depression with a maximum effect at 30 minutes (112 ± 6%) post-WBPA. The protective effect of WBPA lasted for three hours. In the presence of L-NAME, WBPA had no effect.

## 4.0 Discussion

This study provides the first evidence that WBPA stimulates TMV as an *in vivo* marker of MCC. This effect was blocked by L-NAME indicating NO dependence. It also demonstrates that blood borne NO and its metabolites can access airway epithelium. WBPA reversed HNE-dependent slowing of TMV, comparable to responses for both experimental pharmacological agents and clinically available drugs.

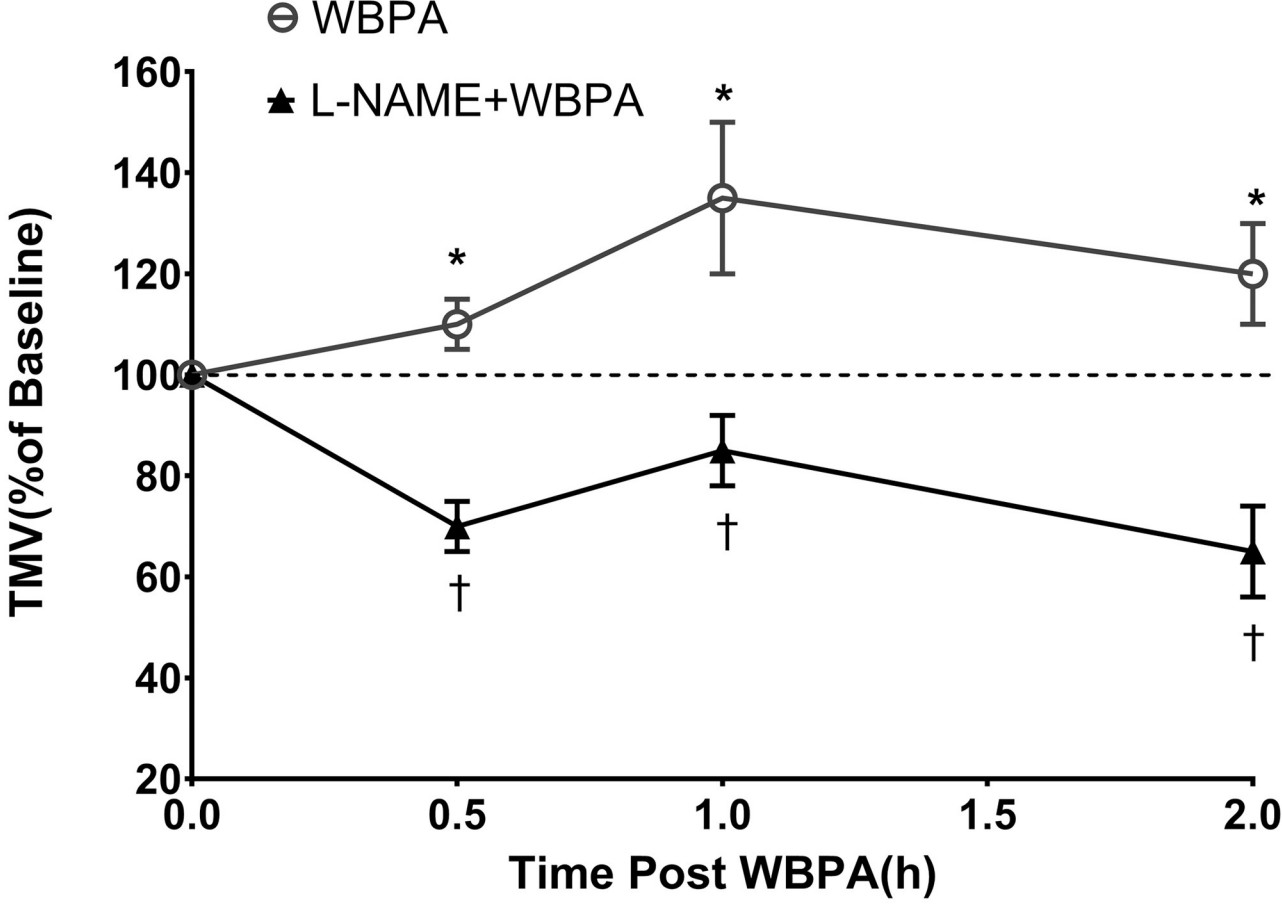

**Fig 1. Changes in Tracheal Mucous Velocity with WBPA.** Changes in tracheal mucus velocity (TMV), expressed as a percentage of baseline, after treatment with 1 hour of WBPA (n = 5) alone and WBPA after treatment with L-NAME before WBPA (n = 5). There was significant increase of TMV (*p < 0.005 vs baseline) at 0.5, 1, and 2 hours post WBPA. L-NAME significantly blunted the effect († p< 0.01 WBPA vs L-NAME+WBPA). Values are mean ± SE. (ANOVA).

In the present study, the peak stimulatory effect of WBPA on TMV (136% above baseline) in awake sheep is slightly greater than the previously reported increase of TMV (74 to 111% above baseline) observed with aerosolized beta adrenergic agonists, isoproterenol and carbuterol in anesthetized dogs [24]. Our prior studies indicate that neutrophil elastase is involved in the antigen-induced reduction in MCC [13] and that when inhaled, it also slows TMV. Bronchodilators have proven to be partly effective in reversing MCC slowing after elastase inhalation. As seen in Fig 2, WBPA reversed HNE-induced slowing of TMV and also increased the value of TMV above baseline. This result has not been previously observed with any other treatment modality. Consistent with our previous results, this WBPA-effect was completely blocked by L-NAME pre-treatment.

The current study expands previous reported results from our laboratory of the diminution of airway resistance in allergic sheep with WBPA. It had been demonstrated that 1-hour of pre-treatment with WBPA protects against allergen-induced bronchoconstriction in allergic sheep, probably mediated by the activity of NO in the regulation of mast cell activation. Further, serial treatments with WBPA over four days protected sheep from developing airway hyper responsiveness after an antigen challenge. [19]

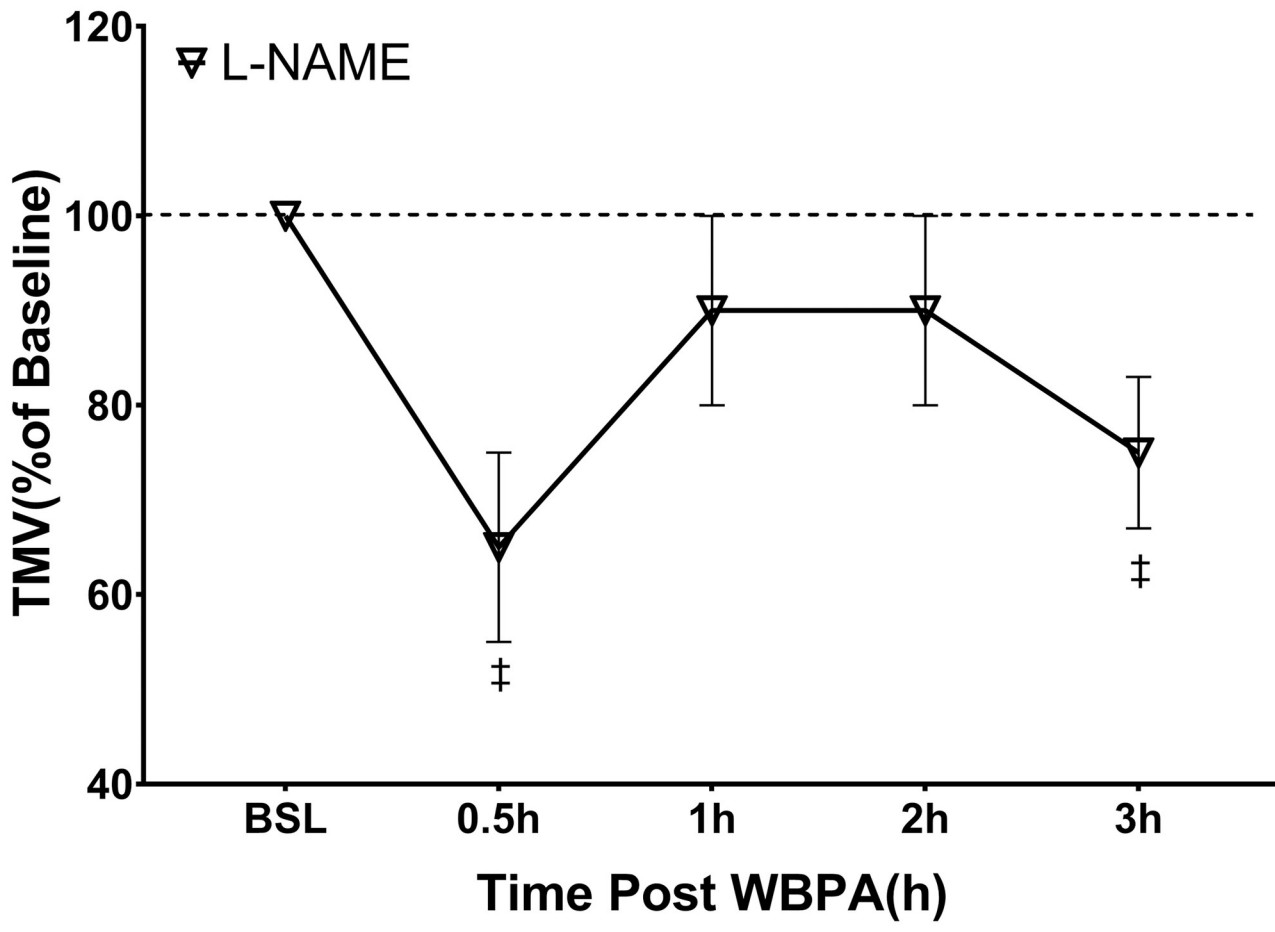

**Fig 2. Change in Tracheal Mucous Velocity Due to NO Inhibition.** Changes in TMV after the administration of L-NAME, 25 mg/kg intravenously (n = 5). Significant difference (‡p<0.05) was present between baseline vs. 0.5 and 3 hours after administration. Although the 1h and 2h mean values were below the initial starting TMV, these values did not differ statistically from the baseline values. Values are mean ± SE (ANOVA).

WBPA has beneficial effects on the diverse manifestations of airway inflammation that produce slowed MCC, bronchoconstriction and airway hyper responsiveness by increasing NO bioavailability. The impact on MCC of serial WBPA treatments in humans and the combination therapy of an inhaled beta agonist with WBPA remain to be investigated.

Bronchopulmonary colonization by *Pseudomonas aeruginosa* causes persistent morbidity and mortality in cystic fibrosis (CF). Chronic *P. aeruginosa* infection in the CF lung is associated with antibiotic-tolerant bacterial aggregates known as biofilms. Disruptions of biofilms have potential to overcome biofilm-associated antibiotic tolerance in CF and other biofilm-related diseases. [25, 26] Submicromolar NO concentrations alone disrupted biofilms within CF sputum and significantly decreased ex vivo biofilm tolerance to tobramycin and tobramycin combined with ceftazidime. In a small randomized clinical trial, 10 ppm NO inhalation significantly reduced *P. aeruginosa* biofilm aggregates compared with placebo across one week treatment.[25] Therefore, low dose NO (nMol/L) achieved with WBPA has the potential to serve as adjunctive therapy in CF.

As limitations to the present study is that we have not characterized whether or not longer or shorter durations of WBPA, can confer greater or less benefits respectively. The action of WBPA releasing NO into circulation as measured by descent of the dicrotic notch of the

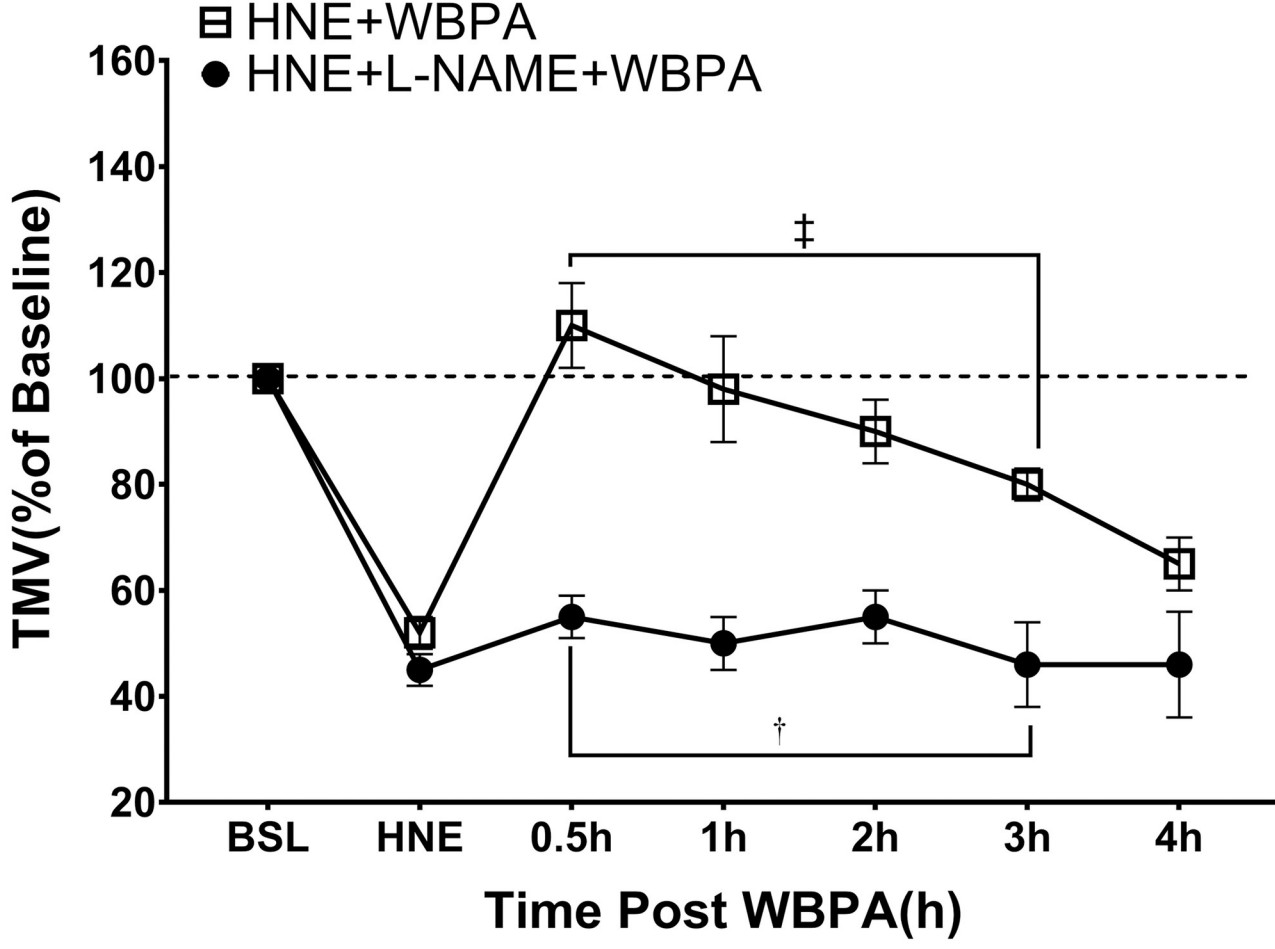

**Fig 3. Change in Tracheal Mucous Velocity after Aerosolized human neutrophil elastase and treatment with WBPA.** Changes in TMV after aerosolized human neutrophil elastase (HNE), and subsequent treatment with either WBPA (n = 5) or L-NAME followed by WBPA (n = 5). There was a significant increase of TMV (‡ p<0.001) at 0.5, 1, 2, and 3 hours post WBPA compared to HNE. L-NAME significantly blunted the effect († p< 0.001 HNE-WBPA vs HNE+L-NAME+WBPA). Values are mean ± SE. (ANOVA).

human finger pulse begins within 15 to 25 seconds after beginning WBPA. [18] The duration of NO action is complicated because NO rapidly binds to circulating proteins and is slowly released to the body in a heterogeneous manner. The current study showed that WBPA stimulates TMV as well as protects against HNE-induced slowing of TMV is in large part due to increase of NO bioavailability. However, the motion platform that produced WBPA has major limitations for human applications. It was too expensive, limited solely to use in the supine posture, and non-portable owing to its large footprint and weight (211Kg). To overcome these limitations, a portable device weighing approximately 5.5 Kg, that can be self-administered was fabricated which is called the GENTLE JOGGER (JD), It incorporates microprocessor controlled, DC motorized movements of foot pedals placed within a chassis to repetitively tap against a semi-rigid surface for simulation of locomotion while the subject is seated or lying in a bed. It is placed on the floor for seated and secured to the footplate of a bed for supine applications. Its foot pedals rapidly and repetitively alternate between right and left pedal movements to actively lift the forefeet upward followed by active downward tapping against a semi-rigid bumper placed within the chassis. In this manner, it simulates feet impacting against the ground during locomotion. Each time the passively moving foot pedals strike the bumper, a

small pulse is added to the circulation as a function of pedal speed ranging from about 120 to 190 steps per minute. This technology produces pulsatile shear stress (friction) to the endothelium that increases release of nitric oxide into the circulation from eNOS. As a human application, JD decreases the rapid onset of increased systolic and diastolic blood pressures associated with human physical inactivity in both supine and seated postures. [27]

We conclude that NO released into the circulation from eNOS via vascular pulsatile shear stress from WBPA acutely accesses the airway epithelium, improving slowed MCC. The latter has significant clinical implications in CF and other inflammatory lung diseases with decreased MCC.

## Supporting information

**S1 Table. The ARRIVE guidelines checklist.** This table contains the ARRIVE guidelines checklist for animal research reporting for in vivo experiments.
(PDF)

## Acknowledgments

We are grateful to the contribution of the late William M Abraham, PhD in study design, critical review of the methodology and data, and his tireless efforts to advance pulmonary physiology and pharmacology.

## Author Contributions

**Conceptualization:** Juan R. Sabater, Marvin A. Sackner, Jose A. Adams, William M. Abraham.

**Data curation:** Juan R. Sabater, Jose A. Adams, William M. Abraham.

**Formal analysis:** Juan R. Sabater, Marvin A. Sackner, Jose A. Adams, William M. Abraham.

**Funding acquisition:** Jose A. Adams, William M. Abraham.

**Investigation:** Juan R. Sabater, Jose A. Adams, William M. Abraham.

**Methodology:** Jose A. Adams, William M. Abraham.

**Project administration:** Juan R. Sabater, Jose A. Adams, William M. Abraham.

**Supervision:** Juan R. Sabater, William M. Abraham.

**Validation:** Marvin A. Sackner, Jose A. Adams.

**Visualization:** Jose A. Adams.

**Writing – original draft:** Juan R. Sabater, Marvin A. Sackner, Jose A. Adams, William M. Abraham.

**Writing – review & editing:** Juan R. Sabater, Marvin A. Sackner, Jose A. Adams.

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
