## [Decision Letter · Decision Letter 0]

25 Sep 2019

PONE-D-19-21481

WHOLE BODY PERIODIC ACCELERATION IN NORMAL AND REDUCED MUCOCILIARY CLEARANCE OF CONSCIOUS SHEEP

PLOS ONE

Dear Dr. Adams,

Thank you for submitting your manuscript to PLOS ONE. After careful consideration, we feel that it has merit but does not fully meet PLOS ONE’s publication criteria as it currently stands. Therefore, we invite you to submit a revised version of the manuscript that addresses the points raised during the review process.

We would appreciate receiving your revised manuscript by Nov 09 2019 11:59PM. To enhance the reproducibility of your results, we recommend that if applicable you deposit your laboratory protocols in protocols.io, where a protocol can be assigned its own identifier (DOI) such that it can be cited independently in the future. For instructions see: http://journals.plos.org/plosone/s/submission-guidelines#loc-laboratory-protocols

We look forward to receiving your revised manuscript.

Kind regards,

Jerome W Breslin, PhD

Academic Editor

PLOS ONE

Journal Requirements:

1. As part of your revision, please complete and submit a copy of the ARRIVE Guidelines checklist, a document that aims to improve experimental reporting and reproducibility of animal studies for purposes of post-publication data analysis and reproducibility: https://www.nc3rs.org.uk/arrive-guidelines. Please include your completed checklist as a Supporting Information file. Note that if your paper is accepted for publication, this checklist will be published as part of your article. Specifically, please include the approval number given to your study by your ethics committee; the source of the animals used in your study; methods of anesthesia and/or analgesia; efforts to alleviate suffering; and what the disposition of the animals was at the end of the study (eg. method of euthanasia, were returned to the herd).

2. Thank you for including your competing interests statement; "I have read the journal's policy and the authors of this manuscript have the following competing interests:

The following Authors SJ,AW have declared that no competing interests exist JAA owns minimal number of stocks in Noninvasive Monitoring Systems (NIMS), a company which manufactures a platform similar to the one described in this study, and co-patent owner of Gentle Jogger. MAS is president of Sackner Wellness Products, a company which manufactures a wellness device called Gentle Jogger referenced in the manuscript, and co-patent owner. This does not alter our adherence to PLoS OnE policies on sharing data and materials. "

We note that you have a patent relating to material pertinent to this article. Please provide an amended statement of Competing Interests to declare this patent (with details including name and number), along with any other relevant declarations relating to employment, consultancy, patents, products in development or modified products etc. Please confirm that this does not alter your adherence to all PLOS ONE policies on sharing data and materials, as detailed online in our guide for authors http://journals.plos.org/plosone/s/competing-interests by including the following statement: "This does not alter our adherence to  PLOS ONE policies on sharing data and materials.” If there are restrictions on sharing of data and/or materials, please state these. Please note that we cannot proceed with consideration of your article until this information has been declared.

Reviewers' comments:

Reviewer's Responses to Questions

**Comments to the Author**

1. Is the manuscript technically sound, and do the data support the conclusions?

Reviewer #1: Yes

Reviewer #2: Yes

2. Has the statistical analysis been performed appropriately and rigorously? 

Reviewer #1: Yes

Reviewer #2: Yes

3. Have the authors made all data underlying the findings in their manuscript fully available?

Reviewer #1: Yes

Reviewer #2: Yes

4. Is the manuscript presented in an intelligible fashion and written in standard English?

Reviewer #1: Yes

Reviewer #2: Yes

5. Review Comments to the Author

Reviewer #1: The study measured tracheal mucus velocity (TMV), a marker of mucociliary clearance (MCC) using whole body periodic acceleration (WBPA). TMV measurements were determined before (basal) and after WBPA at different time points. L-NAME was used to rule out the possibility of nitric oxide (NO) generation and its involvement in WBPA. Authors also studied whether WBPA could reverse the decrease in TMV caused by inhaled human neutrophil elastase (HNE), a model of abnormal MCC. L-NAME also modified HNE-induced depression in TMV.

The findings of the present study support the hypothesis and confirm previous work demonstrating that WBPA can stimulate NOS activity and NO production. Manuscript is well written and interesting. Methods section is clear and explained in detail. The authors appropriately cite past literature with similar findings/experimental conditions to theirs. They have also tried to put their findings into a clinical context.

A few issues, however, need to be addressed:

- Specific statistical test used for each figure should be included in the figure legend.

- Tracheal mucus velocity (TMV) was used in the current study as an index of MCC. How changes in TMV correlate with whole-lung clearance? Please explain.

- Fig. 1 shows TMV modulation by WPBA-mediated NO production over a 2 h period post-WBPA. Did the authors try to get data for 2 other additional hours (time point 4 h)? Since there is a decrease in TMV values from 1 to 2 h, I am wondering if values would continue decreasing over time which might indicate a lack of NO production due to substrate depletion (decreasing NO bioavailability).

- Experimental plan shown in Figs 3&4 was prolonged for 4 hours. Were there any signs of irritation and/or drying of the airway from the beginning to the end of the experiments?

- Authors conclude (page 13, line 282) that NO released into the circulation from eNOS via pulsatile flow as a consequence of WBPA accesses the airway epithelium and improves MCC; this is based on pharmacological inhibition with L-NAME (a broad NO inhibitor) rather than direct eNO measurements. Is there any evidence for iNOS or nNOS involvement under these experimental conditions?

Reviewer #2: The manuscript by Sabater et al. describes the use of whole body periodic acceleration (WBPA) to enhance endothelial nitric oxide (NO) production which in turn increases mucociliary clearance (MCC). This approach can potentially be of significant benefit to patients with cystic fibrosis and other inflammatory lung diseases as a means of enhancing host defense against pathogens. A model of cystic fibrosis in sheep was established via administration of aerosolized human neutrophil elastase (HNE) which suppresses tracheal mucus velocity (TMV) which is a reflection of mucociliary clearance (MCC). The model established represents an acute model of lung inflammation and is appropriate for this study.

The significance of this work is the fact that other nitric oxide (NO) donors drugs increase ciliary beat frequency of

nasal epithelium without increasing mucociliary clearance but WPA clearly increases MCC. This is possibly due to the fact that the NO release is more significant as WPA increases endothelial NO throughout the body. Mechanistically, they also clearly show that this is a NO dependent phenomenon and L-NAME blunts this effect. Overall, this is a well designed study that clearly shows benefit from a simple non-invasive technique to temporarily increase NO levels and NO released systemically impacts the airways.

Concerns relate to the duration of WPA and the lack of discussion regarding shorter/longer duration. Can you quantify, systemic increase in NO levels? Adding discussion regarding why this technique outperforms other NO donor drugs may also help. Also,wouldn't continual administration of HNE represent a situation similar to that seen in inflammatory lung diseases?

6. PLOS authors have the option to publish the peer review history of their article (what does this mean?). If published, this will include your full peer review and any attached files.

Reviewer #1: No

Reviewer #2: Yes: Palaniappan Sethu

---

## [Author Response · Author response to Decision Letter 0]

2 Oct 2019

Response to reviewers

Reviewer #1: 

We thank the reviewer for his/her laudatory comments on the scientific merit, content and writing of the manuscript and the acknowledgement of clinical utility.

A few issues, however, need to be addressed:

- Specific statistical test used for each figure should be included in the figure legend.

 For an unknown reason, these were not uploaded to the manuscript submitted, they are now included in the legend.

- Tracheal mucus velocity (TMV) was used in the current study as an index of MCC. How changes in TMV correlate with whole-lung clearance? Please explain.

TMV reflects changes in whole lung clearance measured with radioactive labeled human serum albumin.(1, 2)

- Fig. 1 shows TMV modulation by WPBA-mediated NO production over a 2 h period post-WBPA. Did the authors try to get data for 2 other additional hours (time point 4 h)? Since there is a decrease in TMV values from 1 to 2 h, I am wondering if values would continue decreasing over time which might indicate a lack of NO production due to substrate depletion (decreasing NO bioavailability).

Low arginine bioavailability plays a pivotal role in the pathogenesis of a growing number of varied diseases, including sickle cell disease, thalassemia, malaria, acute asthma, cystic fibrosis, pulmonary hypertension, cardiovascular disease, certain cancers, and trauma, among others. Catabolism of arginine by arginase enzymes is the most common cause of an acquired arginine deficiency syndrome, frequently contributing to endothelial dysfunction and/or T‐cell dysfunction, depending on the clinical scenario and disease state. (3) None of these factors would be expected in healthy sheep. However, the duration of effects on TMV requires further study and was not addressed.

- Experimental plan shown in Figs 3&4 was prolonged for 4 hours. Were there any signs of irritation and/or drying of the airway from the beginning to the end of the experiments?

There were no obvious signs of airway irritation (coughing or hypersecretion) observed throughout the study. In order to reduce the effects of airway desiccation the animal was placed on room temperature humidify air between measurements

- Authors conclude (page 13, line 282) that NO released into the circulation from eNOS via pulsatile flow as a consequence of WBPA accesses the airway epithelium and improves MCC; this is based on pharmacological inhibition with L-NAME (a broad NO inhibitor) rather than direct eNO measurements. Is there any evidence for iNOS or nNOS involvement under these experimental conditions?

 As mentioned by reviewer L-NAME also inhibits nNOS which is constitutively present in the mucous cells of the epithelium. (4) In rats. eNOS is phosphorylated at Ser-1177. eNOS phosphorylation (Ser-1177) was increased to 261% of control by 1 h of WBPA (360 cpm). Compared to controls, a single 1-h exposure to WBPA (360 cpm) increased the protein level of eNOS in the heart by 393 ± 85% and 461 ± 78% at 4 and 24 h, respectively. The increases of nNOS were 167 ± 9% and 189 ± 36% at 4 and 24 h, respectively. Consistent with these results, mRNA levels of eNOS and nNOS were also significantly increased by WBPA at a frequency of 600 cpm which caused a more sustained increase of eNOS at 4 and 24 h, 392 ± 46% and 534 ± 57% but significantly less induction of nNOS levels (135 ± 19% at 4 h and no change at 24 h compared to WBPA at a frequency of 360 cpm. Inducible nitric oxide synthase (iNOS) was not detected in control or with WBPA at either frequency. Moreover, we did not measure nNOS in airway epithelium with administration of WBPA. 

Reviewer #2: 

We thank the reviewer for their acknowledgement on our study design and clarity of our findings. 

The significance of this work is the fact that other nitric oxide (NO) donors drugs increase ciliary beat frequency of nasal epithelium without increasing mucociliary clearance but WPA clearly increases MCC. This is possibly due to the fact that the NO release is more significant as WPA increases endothelial NO throughout the body. Mechanistically, they also clearly show that this is a NO dependent phenomenon and L-NAME blunts this effect. Overall, this is a well-designed study that clearly shows benefit from a simple non-invasive technique to temporarily increase NO levels and NO released systemically impacts the airways.

Concerns relate to the duration of WPA and the lack of discussion regarding shorter/longer duration. Can you quantify systemic increase in NO levels? Adding discussion regarding why this technique outperforms other NO donor drugs may also help. 

The action of WBPA releasing NO into circulation as measured by descent of the dicrotic notch of finger pulse of humans begins within 15 to 25 seconds after beginning WBPA. (5) The duration of NO action is complicated because NO rapidly binds to circulating proteins and is slowly released to the body in a heterogeneous manner. The effects of 1 hr of WBPA on eNOS expression and phosphorylation have been previously published in rats and mice (6, 7)

We could not find any references to NO donor drugs on MCC in airways and cannot respond to question as to difference in performance. Runer and Lindberg showed increases in ciliary beating frequencies with nitroprusside on human nasal mucosa. (8) 

We could not quantify acute systemic increases of NO as reflected by others using serum nitrite measurements because of inconsistent results present in our attempts to make this measurement.

Also, wouldn't continual administration of HNE represent a situation similar to that seen in inflammatory lung diseases?

This is an interesting observation and a reasonable possibility; however we have not attempted to do so, since our sheep model is not a terminal model. Prolonged administration of HNE, may in fact produce a very severe inflammatory lung disease from which there is no survival 

Reference

1. Hirsh AJ, Zhang J, Zamurs A, Fleegle J, Thelin WR, Caldwell RA, et al. Pharmacological properties of N-(3,5-diamino-6-chloropyrazine-2-carbonyl)-N'-4-[4-(2,3-dihydroxypropoxy)phenyl] butyl-guanidine methanesulfonate (552-02), a novel epithelial sodium channel blocker with potential clinical efficacy for cystic fibrosis lung disease. The Journal of pharmacology and experimental therapeutics. 2008;325(1):77-88. doi: 10.1124/jpet.107.130443. PubMed PMID: 18218832.

2. Sabater JR, Mao YM, Shaffer C, James MK, O'Riordan TG, Abraham WM. Aerosolization of P2Y(2)-receptor agonists enhances mucociliary clearance in sheep. J Appl Physiol (1985). 1999;87(6):2191-6. doi: 10.1152/jappl.1999.87.6.2191. PubMed PMID: 10601167.

3. Morris CR, Hamilton-Reeves J, Martindale RG, Sarav M, Ochoa Gautier JB. Acquired Amino Acid Deficiencies: A Focus on Arginine and Glutamine. Nutr Clin Pract. 2017;32(1_suppl):30S-47S. doi: 10.1177/0884533617691250. PubMed PMID: 28388380.

4. Donaldson SH, Bennett WD, Zeman KL, Knowles MR, Tarran R, Boucher RC. Mucus clearance and lung function in cystic fibrosis with hypertonic saline. The New England journal of medicine. 2006;354(3):241-50. doi: 10.1056/NEJMoa043891. PubMed PMID: 16421365.

5. Sackner MA, Gummels E, Adams JA. Nitric oxide is released into circulation with whole-body, periodic acceleration. Chest. 2005;127(1):30-9. doi: 10.1378/chest.127.1.30. PubMed PMID: 15653959.

6. Wu H, Jin Y, Arias J, Bassuk J, Uryash A, Kurlansky P, et al. In vivo upregulation of nitric oxide synthases in healthy rats. Nitric Oxide. 2009;21(1):63-8. doi: 10.1016/j.niox.2009.05.004. PubMed PMID: 19481168; PubMed Central PMCID: PMCPMC3135669.

7. Uryash A, Bassuk J, Kurlansky P, Altamirano F, Lopez JR, Adams JA. Antioxidant Properties of Whole Body Periodic Acceleration (pGz). PLoS One. 2015;10(7):e0131392. doi: 10.1371/journal.pone.0131392. PubMed PMID: 26133377; PubMed Central PMCID: PMCPMC4489838.

8. Runer T, Lindberg S. Effects of nitric oxide on blood flow and mucociliary activity in the human nose. Ann Otol Rhinol Laryngol. 1998;107(1):40-6. doi: 10.1177/000348949810700108. PubMed PMID: 9439387.

---

## [Decision Letter · Decision Letter 1]

22 Oct 2019

WHOLE BODY PERIODIC ACCELERATION IN NORMAL AND REDUCED MUCOCILIARY CLEARANCE OF CONSCIOUS SHEEP

PONE-D-19-21481R1

Dear Dr. Adams,

We are pleased to inform you that your manuscript has been judged scientifically suitable for publication and will be formally accepted for publication once it complies with all outstanding technical requirements.

With kind regards,

Jerome W Breslin, PhD

Academic Editor

PLOS ONE

Additional Editor Comments (optional):

Reviewers' comments:

Reviewer's Responses to Questions

**Comments to the Author**

1. If the authors have adequately addressed your comments raised in a previous round of review and you feel that this manuscript is now acceptable for publication, you may indicate that here to bypass the “Comments to the Author” section, enter your conflict of interest statement in the “Confidential to Editor” section, and submit your "Accept" recommendation.

Reviewer #1: All comments have been addressed

Reviewer #2: All comments have been addressed

2. Is the manuscript technically sound, and do the data support the conclusions?

Reviewer #1: Yes

Reviewer #2: Yes

3. Has the statistical analysis been performed appropriately and rigorously? 

Reviewer #1: Yes

Reviewer #2: Yes

4. Have the authors made all data underlying the findings in their manuscript fully available?

Reviewer #1: Yes

Reviewer #2: Yes

5. Is the manuscript presented in an intelligible fashion and written in standard English?

Reviewer #1: Yes

Reviewer #2: Yes

6. Review Comments to the Author

Reviewer #1: Thank you for taking the time to respond to my questions. Authors have adequately addressed all comments.

Reviewer #2: All previously raised concerns were adequately addressed. The PI has either provided references or clarified via response to the initial review.

7. PLOS authors have the option to publish the peer review history of their article (what does this mean?). If published, this will include your full peer review and any attached files.

Reviewer #1: No

Reviewer #2: Yes: Palaniappan Sethu

---

## [Editor Report · Acceptance letter]

29 Oct 2019

PONE-D-19-21481R1 

Whole Body Periodic Acceleration in Normal and Reduced Mucociliary Clearance of Conscious Sheep. 

Dear Dr. Adams:

I am pleased to inform you that your manuscript has been deemed suitable for publication in PLOS ONE. Congratulations! Your manuscript is now with our production department. 

With kind regards,

on behalf of

Dr. Jerome W Breslin 

Academic Editor

PLOS ONE